# Encorafenib Acts as a Dual-Activity Chemosensitizer through Its Inhibitory Effect on ABCC1 Transporter In Vitro and Ex Vivo

**DOI:** 10.3390/pharmaceutics14122595

**Published:** 2022-11-24

**Authors:** Yu Zhang, Dimitrios Vagiannis, Youssif Budagaga, Ziba Sabet, Ivo Hanke, Tomáš Rozkoš, Jakub Hofman

**Affiliations:** 1Department of Pharmacology and Toxicology, Faculty of Pharmacy in Hradec Králové, Charles University, Akademika Heyrovského 1203, 500 05 Hradec Králové, Czech Republic; 2Department of Cardiac Surgery, Faculty of Medicine, Charles University and University Hospital Hradec Králové, Sokolská 581, 500 05 Hradec Králové, Czech Republic; 3The Fingerland Department of Pathology, Faculty of Medicine, Charles University and University Hospital in Hradec Králové, Sokolská 581, 500 05 Hradec Králové, Czech Republic

**Keywords:** encorafenib, multidrug resistance, ABC transporter, non-small cell lung cancer, cytochrome P450

## Abstract

Encorafenib (LGX818, trade name Braftovi), a novel BRAF inhibitor, has been approved for the treatment of melanoma and colorectal cancer. In the present work, we evaluated encorafenib’s possible antagonistic effects on the pharmacokinetic mechanisms of multidrug resistance (MDR), as well as its perpetrator role in drug interactions. Firstly, encorafenib potently inhibited the efflux function of the ABCC1 transporter in drug accumulation assays, while moderate and null interaction levels were recorded for ABCB1 and ABCG2, respectively. In contrast, the mRNA expression levels of all the tested transporters were not altered by encorafenib. In the drug combination studies, we found that daunorubicin and topotecan resistances were synergistically attenuated by the encorafenib-mediated interaction in A431-ABCC1 cells. Notably, further experiments in ex vivo patient-derived explants confirmed the MDR-modulating ability of encorafenib. Advantageously, the overexpression of tested drug efflux transporters failed to hinder the antiproliferative activity of encorafenib. In addition, no significant modulation of the CYP3A4 enzyme’s activity by encorafenib was observed. In conclusion, our work indicated that encorafenib can act as an effective chemosensitizer targeting the ABCC1-induced MDR. Our in vitro and ex vivo data might provide valuable information for designing the novel effective scheme applicable in the clinical pharmacotherapy of *BRAF*-mutated/ABCC1-expressing tumors.

## 1. Introduction

With a global estimation of 19 million new diagnoses and 10 million deaths every year, cancer is undoubtedly one of the major threats to human health [1]. Although pharmacological interventions serve as an irreplaceable approach to clinical cancer management, their efficacies are often incapacitated by the emergence of multidrug resistance (MDR), which is characterized by the participation of pharmacodynamic and pharmacokinetic mechanisms [2].

Pharmacokinetic factors promoting MDR are mainly associated with the decrease in drugs’ intracellular levels and/or the enhancement of the drug-deactivating process [3,4]. ATP-binding cassette (ABC) transporters are a class of transmembrane proteins that facilitate the efflux of substrates into the extracellular environment, thereby protecting normal tissues from xenobiotics (including drugs). In tumor cells, however, this protective function switches to a factor that reduces cellular responsiveness to therapeutic agents by decreasing their intracellular concentrations [5]. Among 49 identified members, P-glycoprotein (ABCB1), breast cancer resistance protein (ABCG2) and MDR-associated protein 1 (ABCC1) have been well-clarified to play critical roles in the promoting MDR phenomenon in cancer [6,7]. Next to the role in MDR, ABCB1 and ABCG2 are important sites for pharmacologically relevant drug–drug interactions (DDIs). Drug-metabolizing enzymes represent additional ADME entities significantly contributing to MDR and DDIs. Phase I biotransformation reactions are mainly mediated by the enzymes from the cytochrome P450 superfamily (CYPs). CYP3A4, the leading isozyme, catalyzes metabolic transformations of more than 50% of the commonly prescribed clinical agents [8]. Our recent paper also pointed out its role in the cytostatic MDR; we demonstrated that the cytotoxicity and pro-apoptotic properties of docetaxel are abrogated by CYP3A4′s action [9]. The overexpression of ABC transporters (ABCB1, ABCG2, and ABCC1), along with CYP3A4, is frequently observed in various cancer types [5,10]. However, many attempts for the reversal of pharmacokinetic MDR with conventional chemosensitizers have been unsuccessful with low or no benefits to patients, predominantly due to the lack of genotyping, high toxicity, and/or inducing unexpected DDIs [11]. Consequently, novel targeted anticancer drugs with dual activities are expected to be effective against MDR. In addition to their own antineoplastic effects, these targeted agents can also synergistically potentiate the sensitivities of cancer cells to chemotherapeutic agents through interactions with ABC transporters and/or CYPs [12].

*BRAF* gene codes for the BRAF serine/threonine kinase, which regulates the mitogen-activated protein kinase (MAPK) pathway, an important cascade governing cell functions such as proliferation, differentiation, cell survival, and apoptosis. *BRAF* mutations are important drivers of tumorigenesis [13,14]. The small molecule BRAF inhibitor, encorafenib (LGX818, trade name Braftovi; Figure 1), has been approved by the FDA for the combined treatment of unresectable or metastatic melanoma harboring a *BRAF*^V600E^ or ^-V600K^ mutation (in 2018), and *BRAF*^V600E^-mutant metastatic colorectal cancer (in 2020) [15,16]. Moreover, encorafenib has been selected as a promising candidate, which is now undergoing several preclinical and clinical evaluations in the various types of cancers, including non-small cell lung cancer (NSCLC; NCT05195632). Its analog, dabrafenib, has already been approved for NSCLC therapy, thus rationalizing this drug developmental strategy [17].

While encorafenib’s pharmacodynamic activities have been intensively studied, its role in pharmacokinetic DDIs and MDR modulation still awaits elucidation. In our study, we aimed to shed the light on this issue using several in vitro and ex vivo experimental techniques, including accumulation and incubation assays, drug combination studies in transduced models as well as patient-derived explants, and gene/protein expression assessments.

## 2. Materials and Methods

### 2.1. Reagent and Chemicals

Encorafenib and topotecan were purchased from MedChem Express (New Jersey, NJ, USA). Dulbecco’s Modified Eagle Medium (DMEM): Nutrient Mixture F-12, Minimum Essential Medium (MEM), and Opti-MEM were obtained from Gibco BRL Life Technologies (Rockville, MD, USA). Calcein AM and Vivid CYP3A4 Screening Kit were bought from Thermo Fisher Scientific (Waltham, MA, USA). Primary antibodies against human ABCC1 (cat. no. sc-18835) as well as a secondary anti-mouse antibody (cat. no. sc-516102) were obtained from Santa Cruz Biotechnology (Dallas, TX, USA). Mouse monoclonal anti-cytokeratin 18 antibody [C-04] (FITC) (cat. no. ab52459) and anti-β-actin (cat. no. ab8226) were purchased from Abcam (Cambridge, MA, USA). TRI Reagent was purchased from the Molecular Research Center (Cincinnati, OH, USA). ProtoScript^®^ II Reverse Transcriptase and Deoxynucleotide (dNTP) Solution Mix were accessed from New England Biolabs (Ipswich, MA, USA). Oligo(dT) was purchased from Generi Biotech (Hradec Kralove, Czech Republic). TaqMan™ Universal Master Mix II (no UNG) for qRT-PCR assay and TaqMan systems designed for evaluating the gene expression levels of *ABCB1*, *ABCG2*, *ABCC1*, *HPRT1,* and *GAPDH* were purchased from Applied Biosystems Life Technologies (Carlsbad, CA, USA). Collagen I was bought from Corning (Corning, NY, USA). Roswell Park Memorial Institute medium 1640 (RPMI-1640), high-glucose DMEM, daunorubicin, mitoxantrone, ketoconazole, hoechst 33342, bovine serum albumin (BSA), trypsin inhibitor, 3-(4,5-dimethyl-2-thiazolyl)-2,5-diphenyl-2H-tetrazolium bromide (MTT), dimethyl sulfoxide (DMSO), fetal bovine serum (FBS), phosphate-buffered saline (PBS), HEPES solution, sodium pyruvate solution, as well as primary culture chemicals (collagenase, ethanolamine, Ficoll Paque Plus, gentamicin, growth factors, hormones, penicillin/streptomycin, phosphoethanolimine, pituitary extract, and triiodothyronine) were bought from Sigma Aldrich (St. Louis, MO, USA).

### 2.2. Cell Culture

Parent Madin-Darby canine kidney II (MDCKII-par) cell line and its human ABC transporter-overexpressing counterparts (MDCKII-ABCB1, MDCKII-ABCG2, and MDCKII-ABCC1) were obtained from Dr. Alfred Schinkel (The Netherlands Cancer Institute, Amsterdam, The Netherlands). Parent human squamous carcinoma A431 cells and their human ABC transporter (ABCB1, ABCG2, and ABCC1)-transduced sublines were accessed from Dr. Balasz Sarkadi (Hungarian Academy of Sciences, Budapest, Hungary). NCI-H2228 and HCC827 (human NSCLC cell lines) were obtained from the American Type Culture Collection (Manassas, VA, USA). The cells were cultured under a 5% CO_2_ atmosphere at 37 °C (the mycoplasma infection tests were periodically performed). For the experiments, cells from passages 10–25 were used. High-glucose DMEM containing 10% FBS was applied for the maintenance of MDCKII and A431 cells, while RPMI-1640 supplemented with 10% FBS, 10 mM HEPES and 1 mM sodium pyruvate was used for the cultivation of NCI-H2228 and HCC827 cells. Encorafenib and some other compounds were dissolved in DMSO; its concentration was kept lower than 0.5% for all experiments. Relevant vehicle controls were employed for the elimination of possible influences of DMSO on the measurement outcomes.

### 2.3. Isolation and Purification of NSCLC Patient-Derived Primary Cells

Tumor biopsies taken from NSCLC patients were generously provided by the Department of Cardiac Surgery, University Hospital Hradec Kralove. The text of the informed consent was approved by the University Hospital Ethics Committee (study no. 202002 S04P) and was signed by all participants. In this study, samples 1, 2, 3, 4, 5, and 6 were obtained from adenocarcinoma (male, 73 years old), squamous carcinoma (female, 67 years old), adenocarcinoma (male, 72 years old), adenocarcinoma (female, 69 years old), combined neuroendocrine carcinoma (male, 64 years old), squamous carcinoma (male, 70 years old), respectively. After the lung lobectomy, tumor tissue was immediately excised by the pathologist to collect an NSCLC sample. The isolation of lung cancer cells was based on the method reported in published papers with minor optimizations [18,19,20]. Fragmented pieces 2–5 mm in diameter were separated from the tumor samples using a scalpel. Cells were released from small tumor pieces in 1 × MEM (containing 0.1% collagenase) after 30-min incubation in a 37 °C water bath. Subsequently, a 40 μm-sized cell strainer was used to collect the cells into the solution of BSA in MEM. After the centrifugation at 200× *g* for 5 min, the obtained pellet was resuspended in BSA-containing MEM solution. The Ficoll Paque Plus was pipetted into the suspension, which was subsequently centrifuged at 100× *g* for 10 min. Cells were collected from the interfacial area into a c-based media. In the resulting suspension, the residual Ficoll was further removed by centrifugation at 200× *g* for 5 min. Finally, the obtained pellet was carefully resuspended in c-based media and then placed into a cell culture flask pre-coated with collagen I. The culture medium was replaced every 2–3 days until the confluence of cells reached 60–70%. For the removal of fibroblast contamination, anti-fibroblast microbeads (Miltenyi Biotec, Bergisch Gladbach, Germany) were applied according to the manufacturer’s protocol. Possible interferences caused by the physiological cells were eliminated by replacing the initially used media with the media containing 10% FBS (after the cells reached 70% confluence again). Anti-cytokeratin 18 antibody was used for staining the cells and the epithelial origin was further confirmed by using Sony SA3800 spectral cell analyzer (Sony Biotechnology, San Jose, CA, USA). For the whole experimental duration, the number of passages for NSCLC primary culture cells was no more than 4.

### 2.4. Drug Accumulation Study for ABC Transporters

The drug accumulation study was performed following the descriptions in our previous papers [20,21,22]. MDCKII-par, MDCKII-ABCB1, MDCKII-ABCG2, MDCKII-ABCC1, and lung cancer explant cultures were seeded in 12-well plates at the densities of 2.2 × 10^5^, 1.5 × 10^5^, 2.5 × 10^5^, 2.2 × 10^5^, 1.5 × 10^5^ cells/well, respectively. After 24-h cultivation, cells were carefully rinsed with warm 1 × PBS and then exposed to the designed encorafenib’s dilutions or model inhibitors (1 μM LY335979 for ABCB1, 1 μM Ko143 for ABCG2 and 25 μM MK-571 for ABCC1) for 10 min. Fluorescent substrates of ABC transporters (2 μM daunorubicin or 5 μM mitoxantrone) were subsequently added to the cells for a 60-min treatment. Afterwards, plates were placed on ice, and cells were treated with phenol red-free trypsin. The ice-cold 1 × PBS solution containing 2% FBS was added to resuspend the cells. Using a flow cytometer (BD FACSCanto II, Allschwil, Switzerland), substrates’ levels were detected under the excitation/emission wavelengths equal to 490/565 and 640/670 nm for daunorubicin and mitoxantrone, respectively.

### 2.5. MTT Proliferation Assay

Cell viability was evaluated based on the MTT assay as described in our previous papers [19,20]. The cells were seeded in 96-well plates at the following densities for a 24-h incubation: MDCKII-par, MDCKII-ABCB1, MDCKII-ABCG2, and MDCKII-ABCC1: 1.5 × 10^4^ cells/well; A431-par, A431-ABCB1, A431-ABCG2, and A431-ABCC1: 2.0 × 10^4^ cells/well; NCI-H2228: 1.2 × 10^4^ cells/well; HCC827: 4.0 × 10^4^ cells/well; NSCLC primary culture cells: 1.0 × 10^4^ cells/well. After 48-h exposure to several concentrations of single drugs or drug combinations, cells were rinsed with a warm 1 × PBS solution and then incubated for 1 h with freshly prepared MTT solution in Opti-MEM (1 mg/mL) at 37 °C. Subsequently, the MTT solution was discarded, and the formazan crystals were dissolved by DMSO, which was followed by a 10-min incubation. The microplate reader (Infinite M200, Tecan) was used for the determination of the absorbance at wavelengths of 570 nm and 690 nm. For further data processing, background absorbance at 690 nm was deducted from the data obtained at 570 nm. A medium-containing-vehicle or 40% DMSO served as 100% or 0% cell viability controls, respectively.

### 2.6. Drug Combination Study

Drug combination studies in A431 and NSCLC primary culture cells were carried out similarly as in our previous descriptions [19,20,22]. Cells were seeded in 96-well plates at their specific densities as mentioned above. Following 24 h incubation, serially diluted cytotoxic agents (topotecan, daunorubicin, or mitoxantrone) with/without 10 μM encorafenib were added to the cells for a 48-h treatment. Subsequently, an MTT assay was performed, and experimental data were obtained with the help of a microplate reader Tecan Infinite M200. To qualify drug combination effects, the combination index (CI) values were determined by Chou-Talalay assay in a data-analytic software CompuSyn 3.0.1 (ComboSyn Inc., Paramus, NJ, USA). Combination effects were defined as synergistic (CI < 0.9), additive (0.9 < CI < 1.1), or antagonistic (CI > 1.1) [23].

### 2.7. Western Blotting

Western blotting experiments were performed according to the previous description [19,20]. Briefly, proteins from the primary culture samples were isolated when the cells reached full confluency in Petri dishes. After rinsing with cold 1 × PBS solution, cells were harvested in the cell lysis buffer (20 mM Tris, 150 mM NaCl, 12.8 mM EDTA, 1 mM EGTA, 4.2 mM Na-pyrophosphate, 1 mM Na_3_VO_4_, and 10 mL/L Triton; 10 μL/mL protease inhibitor cocktail was supplemented into the buffer before use). Total protein was extracted from the whole cell lysate after the centrifugation (12,000× *g* for 30 min at 4 °C). The Bradford assay reagent was subsequently applied for the measurement of the concentration of obtained total protein. For the separation of protein samples (a total loading amount of 40 μg), 8% SDS-PAGE gel was used. Then, using the Trans-Blot TurboTM Transfer System (Bio-Rad Laboratories, Hercules, CA, USA), separated proteins were transferred to PVDF membranes, which was followed by blocking with TBST buffer (containing 5% non-fat dry milk) for 90 min at room temperature. The specific primary antibodies were diluted in TBST buffer in the following ratio: anti-ABCC1 (1:500) and anti-β-actin (1:10,000), and then applied to the membranes at 4 °C overnight. Afterwards, membranes were washed 3 times with TBST buffer, incubated with HRP-conjugated anti-mouse secondary antibody diluted in TBST buffer (1:2000) for 1 h at room temperature and additionally washed 3 times with TBST buffer. Then, Immobilon Western Chemiluminescent HRP Substrate (EMD Millipore, Billerica, MA, USA) was added to the membranes. The bands were visualized by using the Chemi DocTM MP Imaging System (Bio-Rad Laboratories, Hercules, CA, USA) and quantified by ImageJ software (version 1.46r; National Institutes of Health, Bethesda, MD, USA).

### 2.8. Detection of Gene Expression

A quantitative real-time reverse transcription polymerase chain reaction (qRT-PCR) assay was performed for the detection of gene expression as described in our previous paper [20]. NSCLC cell lines (HCC827 and NCI-H2228) were seeded in 12-well plates at the densities of 8.0 × 10^5^ and 2.0 × 10^5^ cells/well, respectively. After 24 h incubation, the cells were treated with the solution of 1 μM encorafenib or 25 μM rifampicin, which was followed by the 24 h or 48 h incubation. TRI Reagent was used for the extraction of total RNA from cells, while NanoDrop ND-1000 spectrophotometer (American Laboratory Trading, East Lyme, CT, USA) was employed for the quantification of its concentration. Two-step reverse transcription was performed using the T100 Thermal Cycler (Bio-Rad Laboratories, Hercules, CA, USA). First, 5 μM oligo(dT) was added to the RNA sample (1 μg) for hybridization at 65 °C for 5 min. Then, a mastermix (4 μL of 5 × ProtoScript^®^ II Reverse Transcriptase Reaction Buffer, 2 μL of 10 × dithiothreitol, 1 μL of 10 mM dNTP Solution Mix and 0.5 μL of 200 U/μL ProtoScript^®^ II Reverse Transcriptase) was supplied to the sample, and the cDNA synthesis cycle was finished in the thermal cycler (42 °C for 50 min followed by 65 °C for 20 min). Subsequently, qRT-PCR based on Taqman probes was performed in 384-well plates with a total final volume of 10 μL via QuantStudio 6 (Life Technologies, Carlsbad, CA, USA), following the protocol which was provided by the manufacturer of Taqman systems. The mRNA expression levels of transporter genes were evaluated by the 2^−ΔΔCt^ method, while *GAPDH* and *HPRT1* were applied as internal references.

### 2.9. Human Recombinant CYP3A4 Enzyme Inhibition Study

A commercial Vivid™ CYP3A4 screening kit (cat. no. P2857, ThermoFisher Scientific, Waltham, MA, USA) was employed for this assay according to the protocol provided by the manufacturer. Serial dilutions of encorafenib (1, 5, 10, 25, and 50 μM) or 10 μM ketoconazole (served as the positive control fully inhibiting CYP3A4) prepared in supplied buffer were added into a black 96-well plate. Then, a mixture of the CYP3A4 isoform and NADPH regeneration system in buffer was applied to the relevant wells. After pre-incubation for 10 min, the reaction was started by adding the freshly prepared solution containing NADP^+^ and the Vivid CYP3A4 substrate. Fluorescence was immediately detected by the microplate reader Tecan Infinite M200 Pro at 1 min intervals for 1 h. The raw data from 15 min were utilized for further analysis. In this experiment, the possible interference caused by DMSO (the solvent for pre-dilution of encorafenib and ketoconazole) was eliminated by applying a final concentration of DMSO (0.5%) in all encorafenib variants and both 100% and 0% enzyme activity samples.

### 2.10. Statistical Analysis

All the experiments were independently performed at least 3 times. Experimental data were analyzed using GraphPad Prism software version 9.4.1 (GraphPad Software Inc., La Jolla, CA, USA). The *p* values were determined using one-way ANOVA followed by Dunnett’s post hoc test or two-tailed unpaired *t*-test as defined in figure legends. Statistical significance was considered when the values of *p* < 0.05 were assessed.

## 3. Results

### 3.1. Encorafenib Shows Functional Inhibitory Effect on ABCC1 in MDCKII Cells

In the first part, we investigated whether encorafenib has functional inhibitory activities toward ABC transporters in transduced MDCKII cells. Cytostatic drugs were employed as the fluorescent model substrates, namely daunorubicin (for ABCB1 and ABCC1) and mitoxantrone (for ABCG2). Our results showed that encorafenib can moderately increase the accumulation of daunorubicin in the MDCKII-ABCB1 cell line with IC_50_ = 25.1 μM (Figure 2a,b). Noteworthy, a significantly increased levels of intracellular daunorubicin were observed in MDCKII-ABCC1 subline even under the treatment with low concentrations of encorafenib (IC_50_ = 8.63 μM; Figure 2e,f). However, in MDCKII-ABCG2 cells, the efflux of mitoxantrone was not significantly affected by the tested drug within the range of tested concentrations (IC_50_ > 20 μM; Figure 2c,d).

### 3.2. Encorafenib Decreases a Cytostatic Resistance Driven by the ABCC1 Overexpression In Vitro

Next, to investigate whether encorafenib can antagonize ABCC1-mediated MDR by its inhibitory effects, drug combination studies were conducted in the A431-par cell line and its ABCC1-overexpressing subline. A total of 10 μM encorafenib was used for the following experiments with respect to its: (1) sufficient inhibitory effects on ABCC1, (2) negligible cytotoxicity in the chosen cell lines, (3) encorafenib’s C_max_ value (5–22 μM) monitored in the clinical trial for 450 mg dose, which is currently used in the clinical practice [24].

The data obtained from drug combination studies showed that encorafenib can significantly alleviate the resistance of ABCC1-overexpressing A431 cells to daunorubicin (R_R_ = 2.59) and topotecan (R_R_ = 2.51), respectively. On the other hand, encorafenib failed to induce statistically significant shifts of IC_50_s of both cytostatic agents in A431 parental cells (Figure 3a,b, Table 1). To accurately evaluate the drug combination effects, the Chou–Talalay analytic method was employed to calculate CIs from our data. F_A_ (fractions of cells affected)-CI plots were subsequently presented for each of the combination settings. Synergistic effects were observed for the combination of encorafenib with daunorubicin/topotecan along the major range of F_A_ in A431-ABCC1. Nevertheless, in the parental cell line, antagonism and/or additivity were clearly defined in both combination variants (Figure 3c,d). Thus, encorafenib was proven to counteract ABCC1-mediated MDR through the interaction with this efflux transporter.

### 3.3. Encorafenib Is Effective against MDR in NSCLC Primary Explants

Subsequently, we selected six different NSCLC patient-derived primary cultures to evaluate the possible clinical impact of our obtained in vitro data. First, we detected the expression levels of ABCC1 in the selected set of primary tumor explants (Figure 4a,b). Under the treatment of encorafenib or model inhibitor MK571, the increased accumulation levels of daunorubicin and mitoxantrone were found in the primary samples with relatively high expression of ABCC1 (samples 1, 2, 4, and 6). On the contrary, in the samples with low ABCC1 expression (samples 3 and 5), encorafenib and MK571 showed negligible effects in drug accumulation studies (Figure 4c). In the follow-up drug combination assays, daunorubicin and mitoxantrone were further used as model MDR victims of ABCC1. Interestingly, synergistic effects were defined for the major parts of CI-F_A_ plots in the NSCLC primary samples with high levels of ABCC1. In contrast, additive or antagonistic outcomes were mainly observed for the drug combination studies in samples 3 and 5 (Figure 4d). Taken together, our ex vivo data indicate a possible favorable MDR-combating capacity of encorafenib, which might be beneficial for cancer patients genotyped for the high intratumoral levels of ABCC1.

### 3.4. The Overexpression of ABCB1, ABCG2 and ABCC1 Fails to Establish Resistance to Encorafenib

To examine whether increased expression levels of ABC transporters could be a promotive factor for encorafenib resistance, we conducted comparative proliferation studies in the two different cellular models with/without ABCB1, ABCG2, and ABCC1 overexpression. Our data showed that the overexpression of studied ABC transporters in the MDCKII cell line did not cause statistical differences in IC_50_ values of encorafenib compared to the parental variant (Figure 5a). Similar phenomena were observed in A431-par cells and their ABC transporter-overexpressing derivatives under the encorafenib treatment (Figure 5b). In a word, our data suggest that the emergence of encorafenib resistance is unlikely to be induced by the increased expression levels of ABCB1, ABCG2, and ABCC1.

### 3.5. Encorafenib Does Not Alter the Gene Expression Levels of MDR-Causing ABC Transporters

Transcriptional upregulation of ABC transporters can significantly affect the resistance profile of tumor cells and/or interfere with drug combination effects. Therefore, we also investigated whether encorafenib might regulate the expression levels of ABC transporters by performing the qRT-PCR-based induction assay in two different NSCLC cell lines. Gene induction studies were conducted by exposing the tested HCC827 and NCI-H2228 cell lines to 1 μM encorafenib, which is an adequate concentration with respect to the viability data (Figure 6a,c) and C_max_ [24]. Our results revealed that encorafenib failed to induce or inhibit the mRNA expression of target transporter genes, as negligible expression fluctuations did not cross over the boundaries of upregulation (increase by 100%) or downregulation (decrease by 50%) (Figure 6b,d) [25]. Noteworthy, the model inducer, rifampicin, was not clearly effective in the examined cellular models except for the *ABCB1* in NCI-H2228. These data reflect that the short-term induction events for MDR-causing ABC transporters in NSCLC tissues are likely to be rare.

### 3.6. Encorafenib Does Not Suppress the CYP3A4 Enzyme’s Activity

Our previous work showed that the CYP3A4 biotransformation enzyme participates in docetaxel resistance [9]. Thus, in the last part, we also investigated the inhibitory potential of encorafenib to this CYPs isoform to find out, whether it might be the potential modulator of this type of pharmacokinetic resistance. Using Vivid™ CYP3A4 screening kit, we found that encorafenib failed to inhibit the activity of CYP3A4 with the IC_50_ > 50 μM (Figure 7). The model inhibitor of CYP3A4, ketoconazole (10 μM), was chosen as a positive control and was observed to suppress the activity of an enzyme to a full extent.

## 4. Discussion

The emergence of MDR impedes the successful outcomes of pharmacological therapies for tumors. Encorafenib (LGX818, trade name Braftovi) is a novel anticancer agent that has been recently approved for the clinical management of unresectable or metastatic melanoma and metastatic colorectal cancer with *BRAF* mutations [15,16]. Currently, several clinical trials are in progress to evaluate the efficacy of encorafenib in the treatment of a wide range of tumor types, including NSCLC [26]. In this work, we assessed encorafenib’s pharmacokinetic interactions and additionally explored its role in MDR.

Initially, we investigated whether encorafenib can inhibit the drug efflux functions of ABCB1, ABCG2, and ABCC1. The drug produced different degrees of inhibitory effects toward the selected transporters (potent inhibition: ABCC1, moderate inhibition: ABCB1, no significant inhibition: ABCG2). Considering the following encorafenib-associated factors: (1) its C_max_ value (ranging from 5–22 μM) found in clinical studies [24]; (2) its plasma protein binding extent (≈86%) [15]; (3) decision algorithms presented by EMA [25], even the moderate inhibition of ABCB1 may have clinical consequences in the terms of causing systemic DDIs. In contrast to ABCB1 and ABCG2, ABCC1 has been proposed as an important transport protein solely for MDR, not for systemic DDIs [25,27,28]. In the next part of our study, we investigated the possible exploitation of the ABCC1-encorafenib interaction for improving the pharmacodynamic activity of cytostatic ABCC1 substrates. Synergistic effects were observed in the combination of encorafenib with both tested cytostatic agents in ABCC1-overexpressing A431 cells, while additivity to antagonism was seen in parental cells. Encouragingly, our ex vivo experiments with patient-derived NSCLC explants showed that encorafenib can effectively enhance the cytotoxicity of MDR-vulnerable agents depending on ABCC1 functional expression. For a better understanding of encorafenib’s ABCC1 inhibitory effect and associated MDR-reversal activity, please see the graphical abstract. Encorafenib combinations with different targeted agents have been clinically evaluated, while some of them have already reached oncological practice (e.g., with binimetinib and cetuximab) [29,30]. In contrast, only a few studies have investigated the combinations of encorafenib with MDR-sensitive conventional chemotherapeutics so far. Specifically, a recent in vitro study demonstrated that methotrexate sensitizes drug-resistant metastatic melanoma cells to encorafenib [31]. Considering our data and the fact, that methotrexate is a recognized ABCC1 substrate, we propose that the sensitizing relationship between these two drugs can be mutual. In addition, it will be interesting to observe the outcomes of the phase III clinical trial that is currently recruiting colorectal cancer-suffering patients (NCT04607421) and is going to evaluate the combination of encorafenib with ABCC1 substrates (irinotecan, 5-fluorouracil). Moreover, ABCC1 is the most frequently overexpressed ABC transporter in the clinical tissue specimens of NSCLC [32] and colorectal carcinoma [33]. Altogether, these and our data suggest that encorafenib might be a clinically relevant chemosensitizer with a dual activity. So far, only a limited number of approved or clinically evaluated tyrosine kinase inhibitors have been confirmed to show inhibitory effects toward ABCC1 transporter. Considering IC_50_ values, the interacting drugs are highly variable in terms of their affinity to ABCC1. Encouragingly, the functional inhibition of ABCC1 was demonstrated to be translatable into the effective reversal of MDR with the exception of brivanib (Table 2).

Subsequently, we evaluated whether encorafenib possibly could play a victim role in the ABC transporter-mediated resistance. Our in vitro comparative proliferation studies showed that encorafenib’s antiproliferative activity is not diminished by the functional presence of drug efflux transporters. In contrast, it was reported that Abcb1 and Abcg2 were able to affect the brain accumulation and disposition of encorafenib in vivo [40]. The authors of this report used only a single dose and very short time intervals in their experiments, which can eventually lead to conflicting conclusions. Encorafenib is a drug with a relatively highly lipophilic character with an estimated logP ≈ 4.16 (according to ALOGPS version 2.1). In general, highly hydrophilic drugs are more susceptible to drug resistance than highly lipophilic agents due to the overwhelming effect of passive diffusion over the membrane carrier-mediated transport in the latter case [41]. We have previously shown that the time-dependent reaching of concentration equilibrium can result in conflicting outcomes of experiments with short vs. long incubation periods [22]. As encorafenib is commonly administered once daily for a total interval of more than 20 weeks [42], it is likely that our proliferation studies are more relevant concerning the prediction of resistance victim properties than the above-discussed brain penetration studies.

Furthermore, the possible regulation of the ABC transporters’ expressions by encorafenib was investigated. Our qRT-PCR data showed that the short-term encorafenib treatment had a negligible effect on the mRNA levels of selected transporters. The obtained results suggest that encorafenib itself does not induce ABC transporter-mediated MDR. Notably, existing studies have indicated that the expression of ABC transporters can be positively regulated by MAPK signaling. Thus, encorafenib can bear an additional MDR-antagonizing potential originating from its primary pharmacodynamic activity, i.e., a long-term MAPK inhibition [43]. Taken together, these data present encorafenib as an ideal chemosensitizer with multiple modes of action.

Last but not least, drug metabolism catalyzed mainly by CYPs is another essential pharmacokinetic factor that induces MDR and is a site for DDIs [8]. Previously, we have reported that CYP3A4 activity is directly associated with cellular resistance to docetaxel [9]. In addition, we have demonstrated that the novel targeted drug, ensartinib, can act as a dual-activity chemosensitizer targeting CYP3A4-mediated MDR [44]. Thus, in the final part, the potential inhibitory effect of encorafenib toward human CYP3A4 was assessed. However, no such significant effect was found up to 50 μM concentration. To sum up, a negligible likelihood exists for encorafenib to serve as a modulator of CYP3A4-mediated resistance. Recent research demonstrated that encorafenib and binimetinib do not significantly influence each other’s metabolic stability or metabolic disposition when used concomitantly [45].

In summary, encorafenib potently and moderately inhibits ABCC1 and ABCB1, respectively, but does not meaningfully interact with ABCG2 or CYP3A4. The tested drug synergistically enhances the cytotoxic effects of MDR-victim anticancer agents in selected ABCC1-positive models in vitro and ex vivo. Notably, the encorafenib’s antiproliferative effect seems not to be hampered by the presence of ABC transporters. Moreover, the drug does not influence the gene expressions of MDR-related transporters. Thus, our findings provide significant preclinical evidence that encorafenib might act as an effective dual-activity chemosensitizer in the combined regimens, especially in those treating BRAF-positive patients co-expressing ABCC1.

## Figures and Tables

**Figure 1 pharmaceutics-14-02595-f001:**
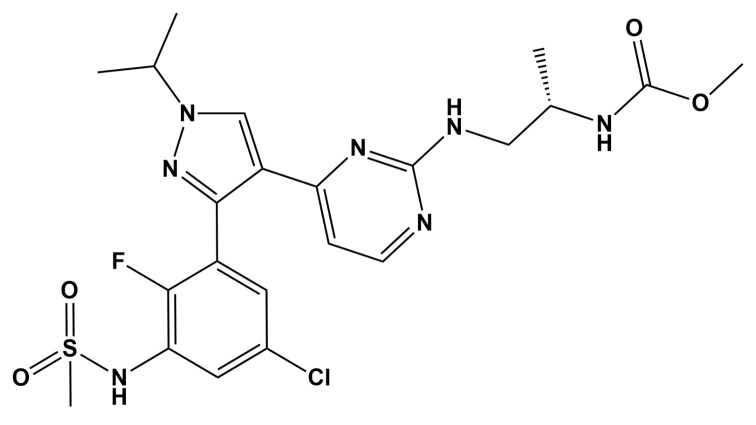
Chemical structure of encorafenib.

**Figure 2 pharmaceutics-14-02595-f002:**
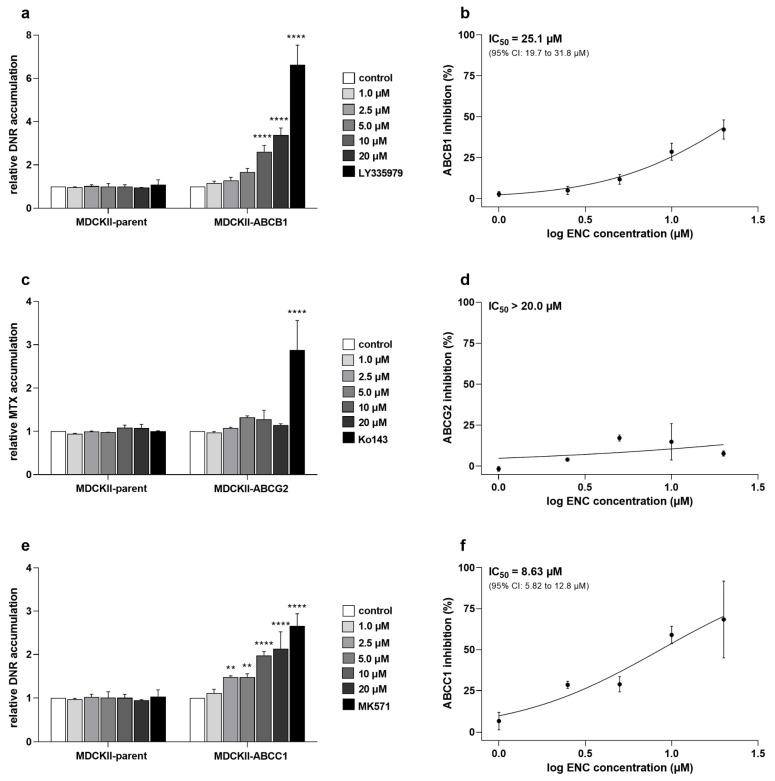
Effects of encorafenib on the accumulation of cytostatic agents in (**a**) MDCKII-ABCB1, (**c**) MDCKII-ABCG2, and (**e**) MDCKII-ABCC1 cells. The tested ABC transporters’ model inhibitors, 1 μM LY335979 (ABCB1), 1 μM Ko143 (ABCG2), and 25 μM MK571 (ABCC1), served as positive controls; their full-inhibiting functions at the selected concentrations were previously confirmed. Cells were firstly pre-incubated with encorafenib/model inhibitors dilutions and then exposed to daunorubicin or mitoxantrone for 60 min. Cells were trypsinized, collected, and analyzed with a flow cytometer. For the processing of obtained data, 0% inhibition and 100% inhibition were represented by vehicle control and model inhibitor values, respectively. After normalization, IC_50_ values were calculated and presented as curves from (**b**) MDCKII-ABCB1, (**d**) MDCKII-ABCG2, and (**f**) MDCKII-ABCC1 cell lines. Statistical analysis was performed using a one-way ANOVA followed by Dunnett’s post hoc test (** *p* < 0.01; **** *p* < 0.0001 vs. control). At least 3 independent experiments were performed. DNR, daunorubicin; ENC, encorafenib; MTX, mitoxantrone; error bar, standard deviation.

**Figure 3 pharmaceutics-14-02595-f003:**
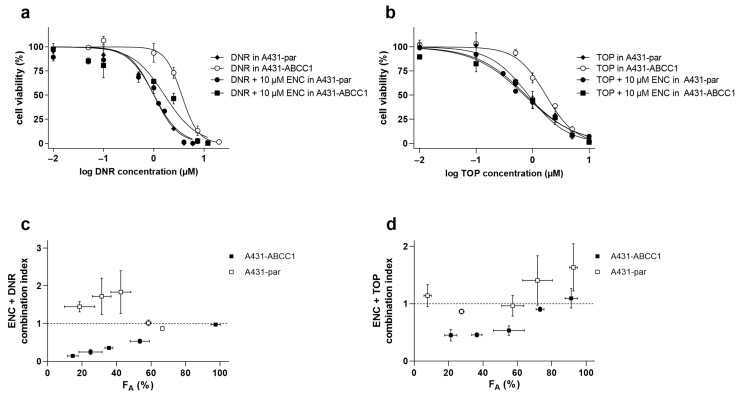
Drug combination effects of 10 μM encorafenib with (**a**) daunorubicin or (**b**) topotecan in ABCC1-overexpressing A431 and respective parental cells. The cells were treated with the cytostatic agents (daunorubicin/topotecan) alone or in combination with 10 μM encorafenib for 48 h. MTT assay was subsequently performed for determining the cell viability. Cell proliferative data further proceeded with the CompuSyn 3.0.1 software, where the combination index (CI) values were calculated. Furthermore, F_A_ (fractions of cells affected)-CI plots were drawn for both combination settings (**c**,**d**). Combination effects were defined according to the CI values as follows: synergistic (CI < 0.9), additive (0.9 < CI < 1.1), or antagonistic (CI > 1.1). At least 3 independent experiments were performed. DNR, daunorubicin; ENC, encorafenib; TOP, topotecan; error bar, standard deviation.

**Figure 4 pharmaceutics-14-02595-f004:**
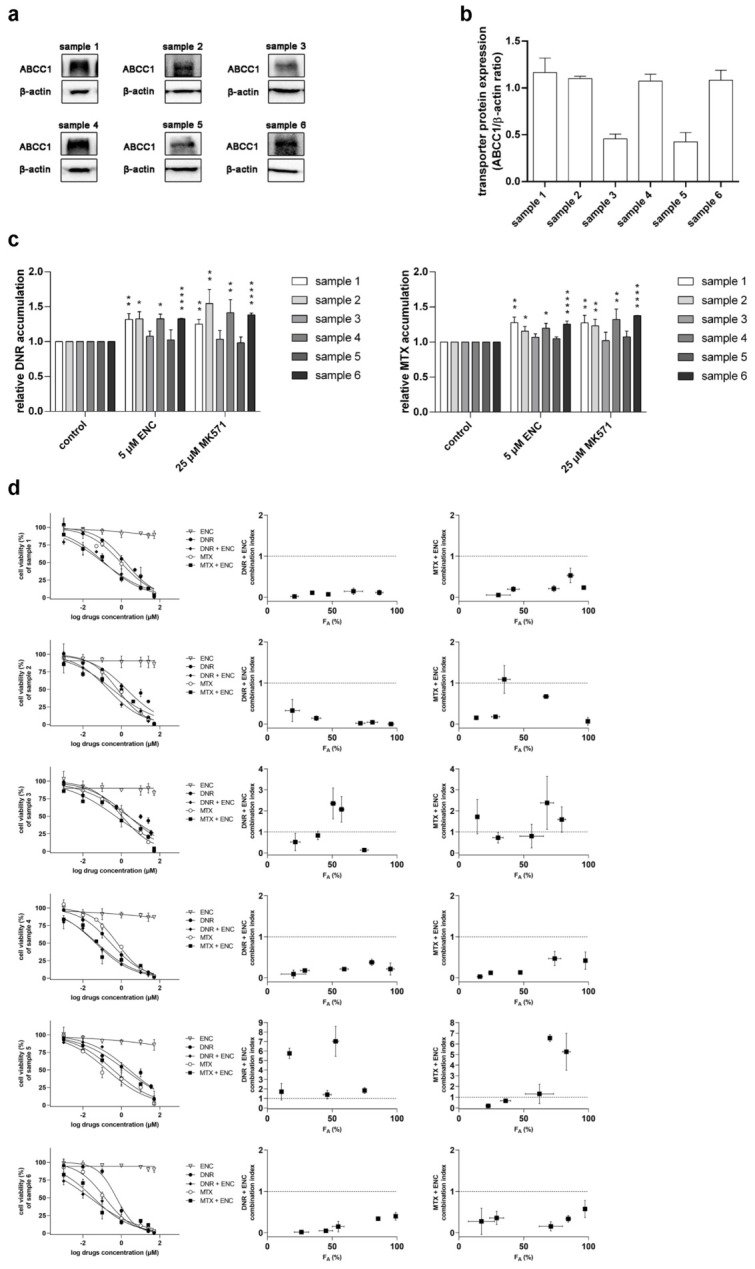
Encorafenib (10 μM) attenuates the intrinsic drug resistance of ex vivo patient-derived NSCLC primary explants to cytostatic MDR victims (daunorubicin and mitoxantrone). First, ABCC1 was detected in primary samples ((**a**) representative pictures from western blotting; (**b**) quantitative densitometric analysis). (**c**) Encorafenib/MK571-induced changes in the accumulation of daunorubicin and mitoxantrone. Significant differences were examined by a one-way ANOVA followed by Dunnett’s post hoc test (* *p* < 0.05; ** *p* < 0.01; **** *p* < 0.0001 vs. control). ((**d**) left) Cytotoxicity of a single drug (encorafenib, daunorubicin, and mitoxantrone) and combined (encorafenib with daunorubicin or mitoxantrone) treatment. ((**d**) right) Combination effects were determined by the Chou-Talalay analysis. F_A_-CI plots are shown. At least 3 experiments were independently performed. DNR, daunorubicin; ENC, encorafenib; MTX, mitoxantrone; error bar, standard deviation.

**Figure 5 pharmaceutics-14-02595-f005:**
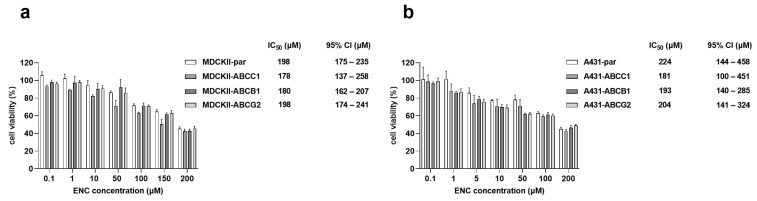
Effect of ABC transporter expression on the reactivity of (**a**) MDCKII and (**b**) A431 cell lines to encorafenib’s antiproliferative activity. Encorafenib was added to the tested cellular models for a 48-h exposure. MTT assay was used for the determination of cell viability. After calculating IC_50_ values, a two-tailed unpaired *t*-test was used for examining the possible significant differences between IC_50_ from parental cell line with IC_50_ from each of the transporter-overexpressing cell lines. As a result, no statistical differences were found in all the comparisons. Each experiment was performed 3 times independently. ENC, encorafenib; error bar, standard deviation.

**Figure 6 pharmaceutics-14-02595-f006:**
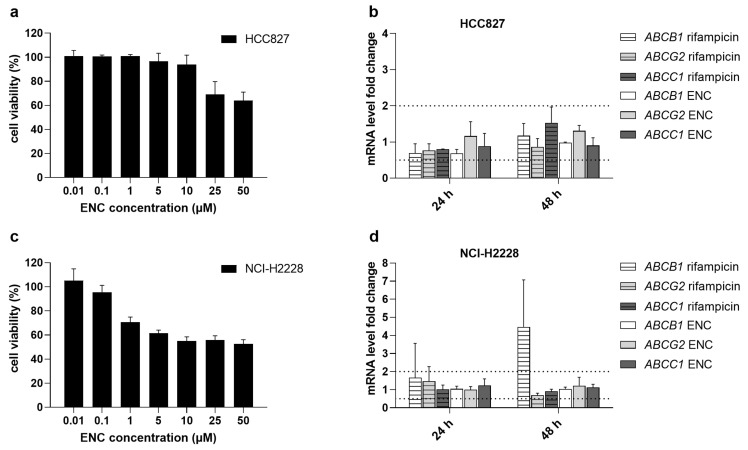
Effects of encorafenib on the mRNA levels of *ABCB1*, *ABCG2,* and *ABCC1* in NSCLC cell lines. Cell viabilities were determined by MTT assay in (**a**) HCC827 and (**c**) NCI-H2228 cells after exposure to encorafenib for 48 h. A total of 1 μM encorafenib or 25 μM rifampicin was added to the tested cells for 1 or 2 days with the daily replacement of drug dilutions. The mRNA expression levels of ABC transporters and house-keeping genes were evaluated by qRT-PCR. After data analysis, fold changes in mRNA levels of examined genes in (**b**) HCC827 and (**d**) NCI-H2228 were presented. According to the drug–drug interaction research guideline released by the European Medicines Agency [25], limits for upregulation or downregulation were defined by the presented dotted lines. At least 3 independent experiments were performed. ENC, encorafenib; error bar, standard deviation.

**Figure 7 pharmaceutics-14-02595-f007:**
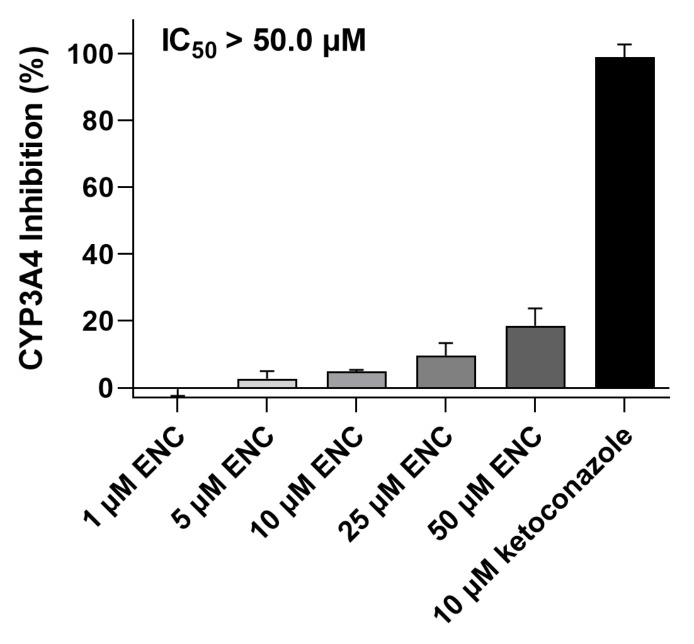
Encorafenib does not induce meaningful inhibitory effects on CYP3A4 isoenzyme’s activity. Experiments were conducted by using a commercial Vivid™ CYP3A4 screening kit according to the provided protocol. For the normalization, raw data from the vehicle control (0.5% DMSO) with/without CYP3A4 enzyme served as the maximal/non-enzyme activity sample. Normalized data were utilized for calculating the IC_50_ value. Experiments were performed as 3 independent repetitions. ENC, encorafenib; error bar, standard deviation.

**Table 1 pharmaceutics-14-02595-t001:** The analysis of IC_50_ shifting for the combination of 10 μM encorafenib with daunorubicin or topotecan in A431-par and A431-ABCC1 cells. Cell viability data from Figure 3 were used for calculating IC_50_ values. Statistics for obtained IC_50_ values was conducted by a two-tailed unpaired *t*-test (** *p* < 0.01; ns, not significant vs. daunorubicin/topotecan alone group). The reversal ratio (R_R_) was calculated for each drug combination in individual cell lines, according to the following formula: R_R_ = IC_50 single drug_/IC_50 drug combination_. DNR, daunorubicin; ENC, encorafenib; TOP, topotecan.

Cell Line	Drug (s)	IC_50_ (μM)	95% CI (μM)	R_R_
A431-par				
	DNR	1.01	0.864–1.18	
	TOP	0.832	0.707–1.01	
	DNR + ENC	1.01 ^ns^	0.900–1.15	1.00
	TOP + ENC	0.764 ^ns^	0.663–0.879	1.09
A431-ABCC1				
	DNR	3.76	3.27–4.00	
	TOP	1.66	1.56–1.90	
	DNR + ENC	1.45 **	1.27–2.13	2.59
	TOP + ENC	0.662 **	0.560–0.911	2.51

**Table 2 pharmaceutics-14-02595-t002:** Summary of ABCC1 inhibitory effects and associated MDR-combating potentials of approved or clinically evaluated (phase II/III) tyrosine kinase inhibitors. NA, not applicable.

Tyrosine Kinase Inhibitor	ABCC1 Inhibition Model	ABCC1 IC_50_(µM)	MDR-Reversal Model	MDR-Victim Cytotoxic Drug	Combination Outcome
encorafenib	accumulation assays in MDCKII cells with daunorubicin	8.63	A431-par and A431-ABCC1 cells	daunorubicin and topotecan	additive to antagonistic effects in A431-par; synergistic effects in A431-ABCC1
alisertib [22]	accumulation assays in MDCKII cells with calcein AM and daunorubicin	19.9 and 2.59, respectively	MDCKII-par and MDCKII-ABCC1 cells	daunorubicin	additivity to antagonism in parent cells; synergism in ABCC1-overexpressing cells
brivanib [34]	accumulation assays in MDCKII cells with calcein AM and daunorubicin	>50.0 and 25.3, respectively	MDCKII-par, MDCKII-ABCC1, A431-par and A431-ABCC1 cells	daunorubicin	additivity to antagonism in all variants
dinaciclib [35]	accumulation assays in MDCKII cells with daunorubicin	18.0	MDCKII-par, MDCKII-ABCC1 and ABCC1-expressing T47D cells	daunorubicin and topotecan	additive to antagonistic effects in MDCKII-par; synergistic effects in MDCKII-ABCC1; weak synergism in T47D
sirolimus [36]	imaging-based accumulation assays in H69 and H69AR cells with calcein AM	2.80	H69AR cells	vincristine	potentiation in resistant H69AR cells
ridaforolimus [36]	imaging-based accumulation assays in H69 and H69AR cells with calcein AM	4.90	H69AR cells	vincristine	potentiation in resistant H69AR cell line
everolimus [36]	imaging-based accumulation assays in H69 and H69AR cells with calcein AM	2.60	H69AR cells	vincristine	potentiation in resistant H69AR cell line
temsirolimus [36]	imaging-based accumulation assays in H69 and H69AR cells with calcein AM	5.60	H69AR cells	vincristine	potentiation in resistant cell line H69AR
ibrutinib [37]	vinblastine accumulation assays in HEK293 cells; doxorubicin accumulation in HL60 cells	NA	HEK293/pcDNA 3.1, HEK293/MRP1, HL60 and HL60/Adr cells	vincristine and doxorobicin	potentiation in resistant cell lines; no effect in parental cells
lapatinib [38]	doxorubicin and rhodamine 123 accumulation assays in KB-3-1 (drug-sensitive) and C-A120 (MRP1 overexpressing) cells	NA	KB-3-1 and C-A120 cells	vincristine and doxorobicin	potentiation in resistant C-A120 cell line; no effect in KB-3-1 drug-sensitive cell line
vandetanib [39]	doxorubicin and rhodamine 123 accumulation assays in KB-3-1 and C-A120	NA	KB-3-1 and C-A120 cells	vincristine	significant IC_50_ shift in resistant C-A120 cell line; non-significant IC_50_ shift in drug-sensitive KB-3-1 cell line

## Data Availability

The authors declare that the data generated and analyzed during this study are included in this published article. In addition, datasets generated and/or analyzed during the current study are available from the corresponding author upon reasonable request.

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
