# Peer review of "Encorafenib Acts as a Dual-Activity Chemosensitizer through Its Inhibitory Effect on ABCC1 Transporter In Vitro and Ex Vivo"

_pharmaceutics, 2022, doi:10.3390/pharmaceutics14122595_

Round 1

Reviewer 1 Report

The manuscript entitled "Encorafenib acts as a dual-activity chemosensitizer through its inhibitory effect on ABCC1 transporter in vitro and ex vivo" deals with the mechanistic evaluation of Encorafenib as chemosensitizer molecule that showed the inhibition of ABCC1 transporter and thereby has potential to deal with the MDR cancer forms. The manuscript overall is written well and proper methods are used to justify the claim. In my view, the manuscript can be accepted for publication in the journal. The only minor change I would suggest is to include a Table and a short paragraph discussing the comparison of IC50 and CI value of Encorafenib with other kinase inhibitors. Also, a figure summarizing its mechanism of inhibition would add more value to the manuscript and would help the readers understand well.

Author Response

The replies to reviewer 1's comments are attached as pdf file.

Reviewer 2 Report

This manuscript could be accepted after minor revision: 

1- Combination of encorafenib and binimetinib should be mentioned.

2- Encorafenib does not suppress CYP3A4 enzyme’s activity 

The following article should be cited indicating that encorafenib is not affecting the metabolism of other drugs (ex binimetinib)

Attwa MW, Darwish HW, Al-Shakliah NS, Kadi AA (2021) A Validated LC–MS/MS Assay for the Simultaneous Quantification of the FDA-Approved Anticancer Mixture (Encorafenib and Binimetinib): Metabolic Stability Estimation. Molecules 26: 2717.

2- Resolution of figures should be increased.

3- In discussion part: a representative figure or a graphical abstract should be included.

Author Response

The replies to reviewer 2's comments are attached as pdf file.
